# Object landmark discovery through unsupervised adaptation

**Enrique Sanchez**[1]

[1] Samsung AI Centre
Cambridge, UK
{e.lozano, georgios.t}@samsung.com

**Georgios Tzimiropoulos**[1,2]

[2] Computer Vision Lab
University of Nottingham, UK
yorgos.tzimiropoulos@nottingham.ac.uk

## Abstract

This paper proposes a method to ease the unsupervised learning of object landmark detectors. Similarly to previous methods, our approach is fully unsupervised in a sense that it does not require or make any use of annotated landmarks for the target object category. Contrary to previous works, we do however assume that a landmark detector, which has already learned a structured representation for a given object category in a fully supervised manner, is available. Under this setting, our main idea boils down to adapting the given pre-trained network to the target object categories in a fully unsupervised manner. To this end, our method uses the pre-trained network as a core which remains frozen and does not get updated during training, and learns, in an unsupervised manner, only a projection matrix to perform the adaptation to the target categories. By building upon an existing structured representation learned in a supervised manner, the optimization problem solved by our method is much more constrained with significantly less parameters to learn which seems to be important for the case of unsupervised learning. We show that our method surpasses fully unsupervised techniques trained from scratch as well as a strong baseline based on fine-tuning, and produces state-of-the-art results on several datasets. Code can be found at `tiny.cc/GitHub-Unsupervised`.

## 1 Introduction

We wish to learn to detect landmarks (also known as keypoints) on examples of a given object category like human and animal faces and bodies, shoes, cars etc. Landmarks are important in object shape perception and help establish correspondence across different viewpoints or different instances of that object category. Landmark detection has been traditionally approached in machine learning using a fully supervised approach: for each object category, a set of pre-defined landmarks are manually annotated on (typically) several thousand object images, and then a neural network is trained to predict these landmarks by minimizing an $L_2$ loss. Thanks to recent advances in training deep neural nets, supervised methods have been shown to produce good results even for the most difficult datasets [2, 1, 37, 22]. This paper attempts to address the more challenging setting which does not assume the existence of manually annotated landmarks (an extremely laborious task), making our approach effortlessly applicable to any object category.

Unsupervised learning of object landmarks is a challenging learning problem for at least 4 reasons: 1) Landmarks are by nature ambiguous; there may exist very different landmark configurations even for simple objects like the human face. 2) Landmarks, although represented by simple x,y coordinates, convey high-level semantic information about objects parts, which is hard to learn without manual supervision. 3) Landmarks must be consistently detected across large changes of viewpoints and appearance. 4) Discovered landmarks must not only be stable with viewpoint change but also fully capture the shape of deformable objects like for the case of the human face and body.

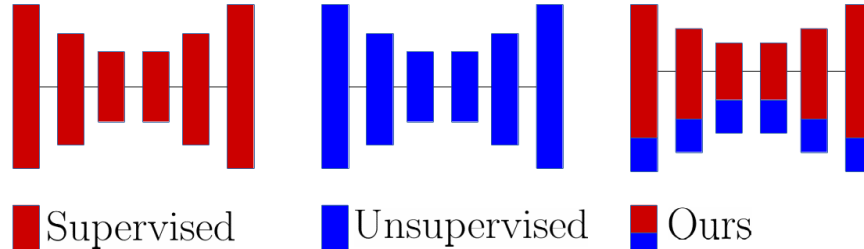

**Figure 1:** Training an object landmark detector: a) Supervised: using heatmap regression, one can learn an object landmark detector from annotated images. b) Unsupervised: using conditional image generation, one can discover the structure of object landmarks. c) Unsupervised, proposed: Our approach uses the "knowledge" learned by a network trained in a supervised way for an object category $\mathcal{X}$ to learn how to discover landmarks for a completely different object category $\mathcal{Y}$ in a fully unsupervised way. Our approach learns only a small fraction of parameters (shown in blue colour) for performing the unsupervised adaptation by solving a more constrained optimization problem which seems to be beneficial for the case of unsupervised learning.

Our method departs from recent methods for unsupervised learning of object landmarks [35, 34, 43, 13] by approaching the problem from an unexplored direction, that of domain adaptation: because landmark annotations do exist for a few object categories, it is natural to attempt to overcome the aforementioned learning challenges from a domain adaptation perspective, in particular, by attempting to adapt a pre-trained landmark detection network, trained on some object category, to a new target object category. Although the pre-trained network is trained in a fully supervised manner, the adaptation to the target object category is done in a fully unsupervised manner. We show that this is feasible by borrowing ideas from incremental learning [28, 25, 26] which, to our knowledge, are applied for the first time for the case of unsupervised learning and dense prediction.

In particular, our method uses the pre-trained network as a core which remains frozen and does not get updated during training, and just learns, in an unsupervised manner, only a projection matrix to perform the adaptation to the target categories. We show that such a simple approach significantly facilitates the process of unsupervised learning, resulting in significantly more accurately localized and stable landmarks compared to unsupervised methods trained entirely from scratch. We also show that our method significantly outperforms a strong baseline based on fine-tuning the pre-trained network on the target domain which we attribute to the fact that the optimization problem solved in our case is much more constrained with less degrees of freedom and significantly less parameters, making our approach more appropriate for the case of learning without labels. As a second advantage, our method adds only a small fraction of parameters to the pre-trained network (about $10\%$) making our approach able to efficiently handle a potentially large number of object categories.

## 2   Related Work

**Unsupervised landmark detection:** While there are some works on learning correspondence [36, 5], or geometric matching [15, 27], or unsupervised learning of representations [33, 38, 6], very little effort has been put towards explicitly building object landmark detectors with no annotations, i.e. in an unsupervised way. While [34] used the concept of *equivariance constraint* [17] to learn image deformations, their proposed approach regresses label heatmaps, that can be later mapped to specific keypoint locations. In order to explicitly learn a low-dimensional representation of the object geometry (i.e. the landmarks), [35] proposed to use a *softargmax* layer [39] which maps the label heatmaps to a vector of $K$ keypoints. The objective in [35] is directly formulated over the produced landmarks, and accounts for the "equivariant error", as well as the diversity of the generated landmarks. Recently, [13] proposed a generative approach, which maps the output of the softargmax layer to a new Gaussian-like set of heatmaps. This combination of softargmax and heatmap generation is referred to as *tight bottleneck*. The new heatmaps are used to reconstruct the input image from a deformed version of it. The bottleneck enforces the landmark detector to discover a set of meaningful and stable points that can be used by the decoder to reconstruct the input image. Concurrent with [13], Zhang *et al.* [43] proposed a generative method that uses an autoencoder to learn the landmark locations. Although their method is based on an encoder-decoder framework, it also relies on the explicit use of both an equivariant and a separation constraints. The method of

[33] learns 3D landmarks detectors from images related by known 3D transformations, again by applying equivariance. Finally, it is worth mentioning other recent works in generative models that, rather than directly predicting a set of landmarks, focus on disentangling shape and appearance in an autoencoder framework [31, 30, 19]. In [31, 30], the learning is formulated as an autoencoder that generates a dense warp map and the appearance in a reference frame. While the warps can be mapped to specific keypoints, their discovery is not the target goal. We note that our method departs from the aforementioned works by approaching the problem from the unexplored direction of unsupervised domain adaptation.

**Incremental learning** combines ideas from multi-task learning, domain adaptation, and transfer learning, where the goal is to learn a set of unrelated tasks in a sequential manner via knowledge sharing [28, 25, 26]. The simplest approach to this learning paradigm is fine-tuning [10], which often results in what is known as "catastrophic forgetting" [7], whereby the network "forgets" a previously learned task when learning a new one. To overcome this limitation, some works have proposed to add knowledge incrementally. An example is the progressive networks [29], which add a submodule, coined adapter, for each new task to be learned. In [25], the adapters are conceived as $1 \times 1$ filters that are applied sequentially to the output features of task-agnostic layers, while in [26], the adapters are applied in a parallel way. An alternative – and equivalent approach – is to directly apply a projection over the different dimensions of the weight tensor. This approach was proposed in [28], where the learnable set of weights reduce to square matrices, which are projected onto the core set of weights before applying the task-specific filters. In this work, we borrow ideas from incremental learning to train a core network in a supervised manner and then adapt it to a completely new object category in a fully unsupervised way. In contrary to all aforementioned methods, our aim, in this work, is not to avoid "catastrophic forgetting" in supervised learning, but to show that such an approach seems to be very beneficial for solving the optimization problems considered in unsupervised learning.

## 3   Method

Our aim is to train a network for landmark detection on data from domain $\mathcal{Y}$, representing an arbitrary object category, without any landmark annotations. We coin this network as the target domain network. To do so, our method firstly trains a network for landmark detection on data from domain $\mathcal{X}$, representing some other object category, in a *supervised* manner. We call this network the core network. The core network is then adapted to give rise to the target domain network in a *fully unsupervised* manner through incremental learning. Note that the core and target networks have the same architecture, in particular, they are both instances of the hourglass network [22] which is the method of choice for supervised landmark localization [22, 3].

**Learning the core network:** Let us denote by $\Psi_{\theta_{\mathcal{X}}}$ the core network, where $\theta_{\mathcal{X}}$ denotes the set of weights of $\Psi$. In a convenient abuse of notation, we will denote $\mathcal{X} = \{\mathbf{x} \in \mathbb{R}^{C \times W \times H}\}$ as the set of training images belonging to a specific object category $\mathcal{X}$ (human poses in our case). For each $\mathbf{x} \in \mathcal{X}$ there is a corresponding set of landmark annotations $\mathbf{a} \in \mathbb{R}^{K \times 2}$, capturing the structured representation of the depicted object. The network $\Psi_{\theta_{\mathcal{X}}}$ is trained to predict the target set of keypoints in unseen images through heatmap regression. In particular, each landmark is represented by a heatmap $\{H_k\}_{k=1,\ldots,K} \in \mathbb{R}^{W_h \times H_h}$, produced by placing a Gaussian function at the corresponding landmark location $\mathbf{a}_k = (u_k, v_k)$, i.e. $H_k(u, v; \mathbf{x}) = \exp(-\sigma^{-2}\|(u, v) - (u_k, v_k)\|^2)$. The network parameters $\theta_{\mathcal{X}}$ are learned by minimizing the mean squared error between the heatmaps produced by the network and the ground-truth heatmaps, i.e. the learning is formulated as:

$$\theta_{\mathcal{X}} = \min_{\theta_{\mathcal{X}}} \sum_{\mathbf{x} \in \mathcal{X}} \|H(\mathbf{x}) - \Psi_{\theta_{\mathcal{X}}}(\mathbf{x})\|^2.$$

For a new image $\mathbf{x}$, the landmarks' locations are estimated by applying the $\arg\max$ operator to the produced heatmaps, that is $\hat{\mathbf{a}} = \arg\max \Psi_{\theta_{\mathcal{X}}}(\mathbf{x})$.

**Learning the target domain network:** Let us denote by $\Psi_{\theta_{\mathcal{Y}}}$ the target domain network, where $\theta_{\mathcal{Y}}$ denotes the set of weights of $\Psi$. Because there are no annotations available for the domain $\mathcal{Y}$, one could use any of the frameworks of [35, 34, 43, 13] to learn $\theta_{\mathcal{Y}}$ in an unsupervised manner from scratch. Instead, we propose to firstly re-parametrize the weights of convolutional layer $\theta_{\mathcal{Y},L}$ as:

$$\theta_{\mathcal{Y},L} = \phi(\mathbf{W}_L, \theta_{\mathcal{X},L}), \tag{1}$$

where $\mathbf{W}_L$ is a projection matrix. The weights $\theta_{\mathcal{X},L}$ are kept frozen i.e. are not updated via back-propagation when training with data from $\mathcal{Y}$. For simplicity, we choose $\phi$ to be the linear function.

Specifically, for the case of convolutional layers, the weights $\theta_{\mathcal{X},L}$ are tensors $\in \mathbb{R}^{C_o \times C_i \times k \times k}$, where $k$ represents the filter size (e.g. $k = 3$), $C_o$ is the number of output channels, and $C_i$ the number of input channels. We propose to learn a set of weights $\mathbf{W}_L \in \mathbb{R}^{C_o \times C_o}$ that are used to map the weights $\theta_{\mathcal{X},L}$ to generate a new set of parameters $\theta_{\mathcal{Y},L} = \mathbf{W}_L \times_1 \theta_{\mathcal{X},L}$, where $\times_n$ refers to the $n$-mode product of tensors. This new set of weights $\theta_{\mathcal{Y},L}$ will have the same dimensionality as $\theta_{\mathcal{X},L}$, and therefore can be directly used within the same hourglass architecture. That is to say, leaving $\theta_{\mathcal{X},L}$ fixed, we learn, for each convolutional layer, a projection matrix $\mathbf{W}_L$ on the number of output channels that is used to map $\theta_{\mathcal{X},L}$ into the set of weights for the target object category $\mathcal{Y}$.

Rather than directly learning $\theta_{\mathcal{Y}}$ as in [13, 43], we propose to learn the projection matrices $\mathbf{W}$ in a fully unsupervised way, through solving the auxiliary task of conditional image generation. In particular, we would like to learn a generator network $\Upsilon$ that takes as input a deformed version $\mathbf{y}'$ of an image $\mathbf{y} \in \mathcal{Y}$, as well as the landmarks produced by $\Psi_{\theta_{\mathcal{Y}}}$, and tries to reconstruct the image $\mathbf{y}$. Specifically, we want our landmark detector $\Psi_{\theta_{\mathcal{Y}}}$ and generator $\Upsilon$ to minimize a reconstruction loss:

$$\min_{\theta_\Upsilon, \mathbf{W}} \mathcal{L}\left(\mathbf{y}, \Upsilon(\mathbf{y}', \Psi_{\theta_{\mathcal{Y}}}(\mathbf{y}))\right).$$

As pointed out in [13], the above formulation does not ensure that the output of $\Psi_{\theta_{\mathcal{Y}}}$ will have the form of heatmaps from which meaningful landmarks can be obtained through the $\arg\max$ operator. To alleviate this, the output of $\Psi_{\theta_{\mathcal{Y}}}$ is firstly converted into $K \times 2$ landmarks, from which a new set of heatmaps is derived using the above mentioned Gaussian-like formulation. The Gaussian heatmaps are then used by the generator $\Upsilon$ to perform the image-to-image translation task. To overcome the non-differentiability of the $\arg\max$ operator, the landmarks are obtained through a *softargmax* operator [39]. This way, the reconstruction loss can be differentiated throughout both $\Upsilon$ and $\Psi_{\theta_{\mathcal{Y}}}$.

Besides facilitating the learning process, our approach offers significant memory savings. For each convolutional layer, the number of parameters to be learned reduces from $C_o \times C_i \times k^2$ to $C_o^2$. For a set-up of $C_i = C_o$ channels, with kernel size $k = 3$, the total number of parameters to train reduces by a factor of 9. The hourglass used in this paper has roughly $\sim 6M$. When using the incremental learning approach described herein, the number of learnable parameters reduces to $\sim 0.5M$.

**Proposed vs. fine-tuning:** An alternative option to our method consists of directly fine-tuning the pre-trained network on the target domain. While fine-tuning improves upon training the network from scratch, we observed that this option is still significantly more prone to producing unstable landmarks than the ones produced by our method. We attribute the improvement obtained by our method to the fact that the core network is not updated during training on domain $\mathcal{Y}$, and hence the optimization problem is much more constrained with less degrees of freedom and with significantly less parameters to learn ($\sim 10\%$ compared to fine-tuning). While fine-tuning has been proven very effective for the case of supervised learning, it turns out that the aforementioned properties of our method make it more appropriate for the case of learning without labels.

**Training:** The training of the network is done using the reconstruction loss defined above. This loss can be differentiated w.r.t. both the parameters of the image encoder-decoder and the projection matrices $\mathbf{W}$. Similarly to [13], we use a reconstruction loss based on a pixel loss and a perceptual loss [14]. The perceptual loss enforces the features of the generated images to be similar to those of the real images when forwarded through a VGG-19 [32] network. It is computed as the $l_1$-norm of the difference between the features $\Phi_{VGG}^l$ computed at layers $l = \{\texttt{relu1\_2}, \texttt{relu2\_2}, \texttt{relu3\_3}, \texttt{relu4\_3}\}$ from the input and generated images. Our total loss is defined as the sum of the pixel reconstruction loss and the perceptual loss:

$$\mathcal{L}(\mathbf{y}, \mathbf{y}') = \|\mathbf{y} - \Upsilon(\mathbf{y}'; \Psi_{\theta_{\mathcal{Y}}}(\mathbf{y}))\|^2 + \sum_l \|\Phi_{VGG}^l(\mathbf{y}) - \Phi_{VGG}^l(\Upsilon(\mathbf{y}'; \Psi_{\theta_{\mathcal{Y}}}(\mathbf{y})))\|.$$

The batch-norm layers are initialized from the learned parameters $\theta_{\mathcal{X}}$ and are fine-tuned through learning the second network on $\mathcal{Y}$. The projection layers are initialized with the identity matrix. Finally, in order to allow the number of points to be (possibly) different for $\mathcal{X}$ and $\mathcal{Y}$, the very last layer of the network, that maps convolutional features to heatmaps, is made domain specific, and trained from scratch.

# 4 Experiments

This section describes the experimental set-up carried out to validate the proposed approach (Sec. 4.1), as well as the obtained results (Sec. 4.2).

## 4.1 Implementation details

**Landmark detector:** It is based on the Hourglass architecture proposed in [22]. It receives an RGB image of size $128 \times 128$, and applies a set of spatial downsampling and residual blocks [8], to produce a set of $K$ heatmaps. Besides the convolutional blocks, the network comprises batch-norm [11] and ReLU layers. The output spatial resolution is $32 \times 32$, which is converted into a $K \times 2$ matrix of coordinates with a softargmax layer ($\beta = 10$). The coordinates are mapped back to heatmaps using $\sigma = \sqrt{0.5}$. In all of our experiments, $K$ is set to 10 points.

**Image encoder-decoder:** The generator is adapted from the architecture used for numerous tasks like neural transfer [14], image-to-image translation [12, 46], and face synthesis [24, 20, 9]. It receives an input image $\mathbf{y}'$ of size $128 \times 128$, and firstly applies two spatial downsampling convolutions, bringing the number of features up to 256. The heatmaps produced by $\Psi_{\theta_\mathcal{Y}}(\mathbf{y})$ are then concatenated to the downsampled tensor, and passed through a set of 6 residual blocks. Finally, two spatial upsampling blocks bring the spatial resolution to the image size.

**Core network pre-training:** For our method, the landmark detector is firstly pre-trained on the task of Human Pose Estimation. In particular, the network is trained to detect $K = 16$ keypoints, corresponding to the human body joints, on the MPII training set [2]. The network is trained for 110 epochs, yielding a validation performance of PCKh = 79%. To study the impact of the quality of the core network on performance (see Sec. 4.2-Additional experiments), we also tested different checkpoints, corresponding to the weights obtained after the 1st, 5th, and 10th epoch of the training process. These models yielded a validation PCKh of 22.95%, 55.03%, and 57.67%, respectively.

**Training:** We generate the pairs $(\mathbf{y}, \mathbf{y}')$ by applying random similarity transformations (scaling, rotation, translation) to the input image. We used the Adam optimizer [16], with $(\beta_1, \beta_2) = (0, 0.9)$, and a batch size of 48 samples. The model is trained for 80 epochs, each consisting of $2,500$ iterations, with a learning rate decay of $0.1$ every 30 epochs. All networks are implemented in PyTorch [23].

**Databases:** For training the object landmark detectors in an unsupervised way, we used the CelebA [18], the UT-Zappos50k [41, 40], and the Cats Head datasets [42]. For CelebA, we excluded the subset of $1,000$ images corresponding to the MAFL dataset [44], and used the remaining $\sim 200$k images for training. For the UT-Zappos50k, we used 49.5k and 500 images to train and test, respectively [35, 43]. Finally, for the Cats Head dataset, we used four subfolders to train the network ($\sim 6,250$ images), and three to test it ($3,750$ images). To perform a quantitative evaluation of our proposed approach, we used the MAFL [44], the AFLW [21], and the LS3D [3] datasets. For MAFL, we used the official train/test partitions. For AFLW, we used the same partitions as in [13]. For LS3D, we used the partitions as defined in [3]. It has to be noted that the LS3D dataset is annotated with 3D points. Similarly to [13], we extracted loose (random) crops around the target objects using the provided annotations. We *did not* use the provided landmarks for training our models.

**Models:** We trained three different models: 1) a network trained directly on each database from *scratch*, 2) a *fine-tuned* network trained by *fine-tuning* the weights of the pre-trained network, and 3) our proposed unsupervised domain adaptation approach. Note that the network trained from scratch is our in-house implementation of [13], while the fine-tuned network is described for the first time in this work, too.

| Method | MAFL | AFLW |
|---|---|---|
| Supervised | | |
| TCDCN [45] | 7.95 | 7.65 |
| MTCNN [44] | 5.39 | 6.90 |
| Unsupervised | | |
| Thewlis [35]($K = 30$) | 7.15 | - |
| Jakab [13]† | 3.32 | 6.99 |
| Jakab [13]†† | **3.19** | 6.86 |
| Zhang [43]($K = 10$) | 3.46 | 7.01 |
| Shu [31] | 5.45 | - |
| Sahasrabudhe [30] | 6.01 | - |
| Ours | | |
| Baseline | 5.00 | 7.65 |
| Finetune | 3.91 | 6.79 |
| Proposed | 3.99 | **6.69** |

**Table 1:** Comparison with state-of-the-art on MAFL and AFLW. For the sake of clarity, we only compare against methods reporting results for $K = 10$ landmarks. †: $K = 10$, uses the VGG-16 for perceptual loss. ††: $K = 10$, uses a pre-trained network for perceptual loss.

|  | $n_{im}$ | MAFL | | | AFLW | | | LS3D | | |
|---|---|---|---|---|---|---|---|---|---|---|
|  |  | Scr. | F.T. | Prop. | Scr. | F.T. | Prop. | Scr. | F.T. | Prop. |
| Forward | 1 | 26.76 | 16.76 | 18.70 | 17.88 | 15.40 | 16.08 | 94.02 | 75.76 | 78.62 |
| | 5 | 18.32 | 9.71 | 8.77 | 16.88 | 13.38 | 12.33 | 70.48 | 45.11 | 43.57 |
| | 10 | 12.12 | 7.45 | 7.13 | 14.62 | 11.59 | 11.09 | 61.31 | 39.26 | 39.37 |
| | 100 | 5.75 | 4.62 | 4.53 | 9.02 | 8.24 | 7.64 | 40.03 | 28.24 | 29.32 |
| | 500 | 5.28 | 4.12 | 4.13 | 8.09 | 7.19 | 7.20 | 34.35 | 25.55 | 27.18 |
| | 1000 | 5.18 | 4.02 | 4.16 | 7.90 | 7.04 | 6.91 | 33.76 | 25.25 | 26.95 |
| | 5000 | 5.04 | 3.98 | 4.05 | 7.67 | 6.81 | 6.73 | 33.25 | 24.75 | 26.50 |
| | All | 5.00 | 3.91 | 3.99 | 7.65 | 6.79 | 6.69 | 33.15 | 24.79 | 26.41 |
| Backward | 1 | 30.64 | 12.50 | 12.30 | 37.23 | 18.92 | 17.47 | 26.11 | 13.96 | 12.31 |
| | 5 | 26.39 | 8.58 | 7.22 | 35.36 | 16.46 | 14.55 | 25.47 | 10.24 | 8.72 |
| | 10 | 22.99 | 7.41 | 6.01 | 32.49 | 15.09 | 12.24 | 20.43 | 8.92 | 7.83 |
| | 100 | 18.86 | 5.23 | 4.23 | 26.36 | 11.72 | 9.69 | 15.25 | 6.32 | 6.08 |
| | 500 | 18.05 | 4.70 | 3.82 | 25.80 | 11.30 | 9.29 | 14.81 | 5.96 | 5.66 |
| | 1000 | 17.82 | 4.60 | 3.74 | 25.60 | 11.23 | 9.25 | 14.59 | 5.91 | 5.55 |
| | 5000 | 17.68 | 4.47 | 3.59 | 25.50 | 11.14 | 9.19 | 14.51 | 5.85 | 5.47 |
| | All | 17.57 | 4.43 | 3.55 | 25.50 | 11.14 | 9.19 | 14.45 | 5.81 | 5.44 |

**Table 2:** Errors on MAFL, AFLW, and LS3D datasets for the *forward* (top), and *backward* (bottom) cases. Scr., F.T., and Prop. stand for Scratch, Fine-tuned, and Proposed, respectively. Despite the good performance of all methods for the forward case, the backward errors clearly show that our method produces the most stable landmarks.

## 4.2 Evaluation

Existing approaches [13, 43, 35] are assessed quantitatively by measuring the error they produce on annotated datasets. To this end, a linear regression is learned from the discovered landmarks and a set of manually annotated points on some training set annotated in the same way as the evaluation dataset. Unfortunately, such a metric does not help measure the stability of the discovered landmarks. In practice, we found that not all discovered landmarks are stable, despite being able to loosely contribute to reducing the error when used to train the regressor.

In order to dig deeper into measuring the stability of our proposed approach, in addition to the aforementioned metric – herein referred to as **forward** - we also measure the error produced by a regressor trained in the reverse order, i.e. from the set of annotated landmarks to the discovered ones. We will refer to this case as **backward**. A method that yields good results in the forward case but poor results in the backward will most likely detect a low number of stable landmarks. Similarly, if a method yields low error in the backward case, but high error in the forward, it will have likely converged to a fixed set of points, independent of the input image. Moreover, we further quantify the stability of landmarks through geometric **consistency**, by measuring the point-to-point distance between a rotated version of the detected landmarks for a given image and those detected on the rotated version of it.

**Forward (Unsupervised → Supervised)**: Following [13, 43, 35], we learn a linear regressor from the discovered landmarks and 5 manually annotated keypoints in the training partitions of MAFL [44] and AFLW [21]. We report the Mean Square Error (MSE), normalized by the inter-ocular distance. Contrary to previous works, we did not re-train our network on AFLW before evaluating it on that dataset. We compare our results against those produced by state-of-the-art methods in Table 1. We can observe that our in-house implementation of [13], although not matching the performance of the original implementation, is competitive ensuring the strength of our baselines and implementations.

The bulk of our results for the forward case are shown in Table 2 (top). Following recent works, we report the results by varying the number of images ($n_{im}$) to train the regressor. For both MAFL and ALFW, we can see that our method surpasses the trained from scratch and fine-tuned networks in all configurations. For LS3D, all methods produce large errors illustrating that there is still a gap to fill in order to make the unsupervised learning of landmark detectors robust to 3D rotations.

**Backward (Supervised → Unsupervised):** For the backward experiment, a regressor is learned from manually annotated keypoints to the landmarks produced by each method. For MAFL and LS3D, this regressor maps the 68 annotated points to the 10 discovered points, while for AFLW the correspondence is from 5 to 10.

Table 2 (bottom) reports the MSE normalized by the inter-ocular distance. As our results show, our method significantly outperforms the fine-tuned network on all datasets and configurations. Especially for MAFL and AFLW datasets, and because the errors for the forward case were also small, these results clearly show that our method produces much more stable points than the fine-tuned network. Note that the trained from scratch network produces very large backward errors indicating that some of the points detected were completely unstable.

**Landmark consistency:** The consistency of the discovered landmarks is quantified by measuring the error per point $e_i = \|\Psi^i_{\theta_{\mathcal{Y}}}(A(\mathbf{y})) - A(\Psi^i_{\theta_{\mathcal{Y}}}(\mathbf{y}))\|$, where $A$ refers to a random similarity transformation. We report the error per

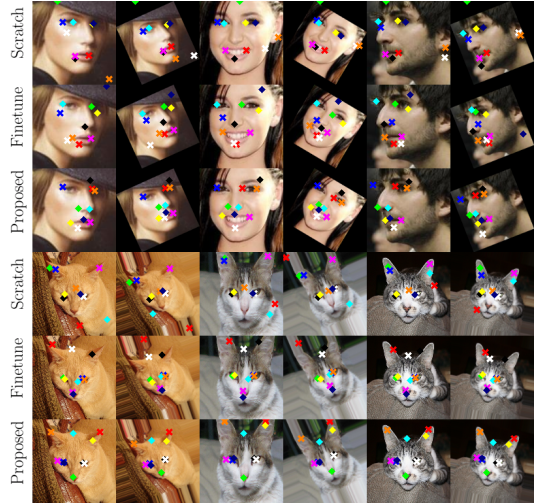

**Figure 2:** Qualitative evaluation of landmark consistency. Each image is transformed using a random similarity transformation (same for each method). Our method consistently produces the most stable points. See paragraph **Landmark consistency** for detailed discussion.

point for each method and dataset in Table 3. Again, it is clear, that our method produces by far the more stable landmarks for all datasets. Qualitative results for this experiment illustrating the stability of the detected landmarks under geometric transformations for all methods can be seen in Fig. 2. It can be seen that, for example for the case of the faces, up to three landmarks are strongly inconsistent for the trained from scratch model (orange, green, and white). For the fine-tuned model, we find two very unstable landmarks (dark blue and white). For our approach, the most unstable landmark is the one depicted in black. However, as shown in Table 3, the error for this landmark is much lower than that of the most unstable landmarks for the fine-tuned and from scratch networks. For the Cats datasets, our approach produces again by far the most stable landmarks.

**Qualitative evaluation:** We show some further examples of the points discovered by our method for all the object categories used in our experiments in Fig. 3. While for the faces and cats datasets, we can draw the same conclusions as above, for the shoes dataset, we observe that all methods showed a consistently good performance (our method is still the most accurate according to Table 3). We attribute this to the fact that the shoes are pre-segmented which turns out to facilitate the training process. However, such a setting is unrealistic for most real-world datasets. More qualitative examples can be found for each dataset in the Supplementary Material.

**Additional experiments:** In addition to the aforementioned experiments, we present two extra studies to illustrate the effect of different training components and design choices in the performance of our proposed approach.

*Quality of core*: Our proposed approach relies on having a strong core network to perform the adaptation. In order to validate this assumption, we repeated the unsupervised training, using as core the saved checkpoints of early epochs of our human pose estimator network (see Sec. 4.1). The forward, backward, and consistency errors for the AFLW database are shown in Table 4 and 5 (Top, # Epoch). The results support the need of having a strong core network for a reliable adaptation.

*Number of training images*: The small amount of parameters to be learned suggests that our unsupervised adaptation method could be robust when training with limited data. To validate this, we chose a random subset of 10, 100, and 1000 images to train the target network. The results of each model are shown in Tables 4 and 5 (Bottom, # Images). When training with 10 images, we observe that the network is prone to collapsing to a single point. However, the results for 1000 images show evidence that our method can be quite effective for the case of limited training data. Combining our approach with few-shot learning is left for interesting future work.

|  |  | 1 | 2 | 3 | 4 | 5 | 6 | 7 | 8 | 9 | 10 | Avg. |
|---|---|---|---|---|---|---|---|---|---|---|---|---|
| MAFL | Scratch | 1.08 | 1.20 | 1.34 | 1.36 | 1.38 | 1.76 | 3.98 | 16.51 | 27.44 | 35.03 | 9.11 |
|  | Finetune | 1.11 | 1.36 | 1.39 | 1.39 | 1.68 | 1.83 | 2.79 | 3.58 | 5.59 | 7.51 | 2.82 |
|  | Proposed | 0.96 | 1.09 | 1.19 | 1.34 | 1.45 | 1.58 | 1.80 | 1.92 | 3.65 | 4.09 | 1.91 |
| AFLW | Scratch | 1.45 | 1.78 | 1.83 | 1.85 | 1.95 | 2.54 | 8.46 | 21.62 | 31.30 | 39.37 | 11.20 |
|  | Finetune | 1.86 | 1.93 | 1.95 | 2.16 | 2.18 | 2.53 | 5.34 | 7.30 | 8.30 | 9.66 | 4.32 |
|  | Proposed | 1.46 | 1.47 | 1.47 | 1.54 | 1.65 | 1.66 | 1.92 | 2.07 | 4.97 | 6.99 | 2.52 |
| LS3D | Scratch | 3.40 | 4.11 | 4.48 | 4.54 | 5.18 | 5.71 | 6.70 | 19.72 | 32.04 | 38.36 | 12.42 |
|  | Finetune | 2.93 | 3.19 | 3.26 | 3.59 | 3.71 | 4.38 | 5.14 | 5.56 | 7.29 | 9.49 | 4.85 |
|  | Proposed | 2.36 | 2.48 | 3.01 | 3.02 | 3.55 | 3.59 | 3.71 | 4.83 | 6.97 | 7.08 | 4.06 |
| Shoes | Scratch | 1.57 | 1.65 | 2.19 | 2.56 | 2.79 | 2.92 | 3.03 | 3.05 | 3.28 | 4.92 | 2.80 |
|  | Finetune | 1.22 | 1.35 | 1.42 | 1.47 | 1.82 | 2.03 | 2.38 | 2.51 | 4.21 | 4.30 | 2.27 |
|  | Proposed | 1.07 | 1.48 | 1.74 | 1.80 | 1.94 | 2.28 | 2.30 | 2.41 | 2.91 | 3.49 | 2.14 |
| Cats | Scratch | 1.27 | 1.44 | 1.61 | 1.82 | 2.30 | 3.37 | 3.46 | 4.44 | 27.13 | 28.11 | 7.50 |
|  | Finetune | 1.27 | 1.48 | 1.81 | 1.82 | 1.82 | 1.84 | 1.89 | 5.48 | 5.93 | 7.14 | 3.05 |
|  | Proposed | 1.00 | 1.01 | 1.25 | 1.60 | 1.65 | 1.79 | 3.57 | 3.60 | 3.64 | 5.29 | 2.44 |

**Table 3:** *Consistency* errors on MAFL, AFLW, LS3D, UT-Zappos50k and Cats Head datasets.

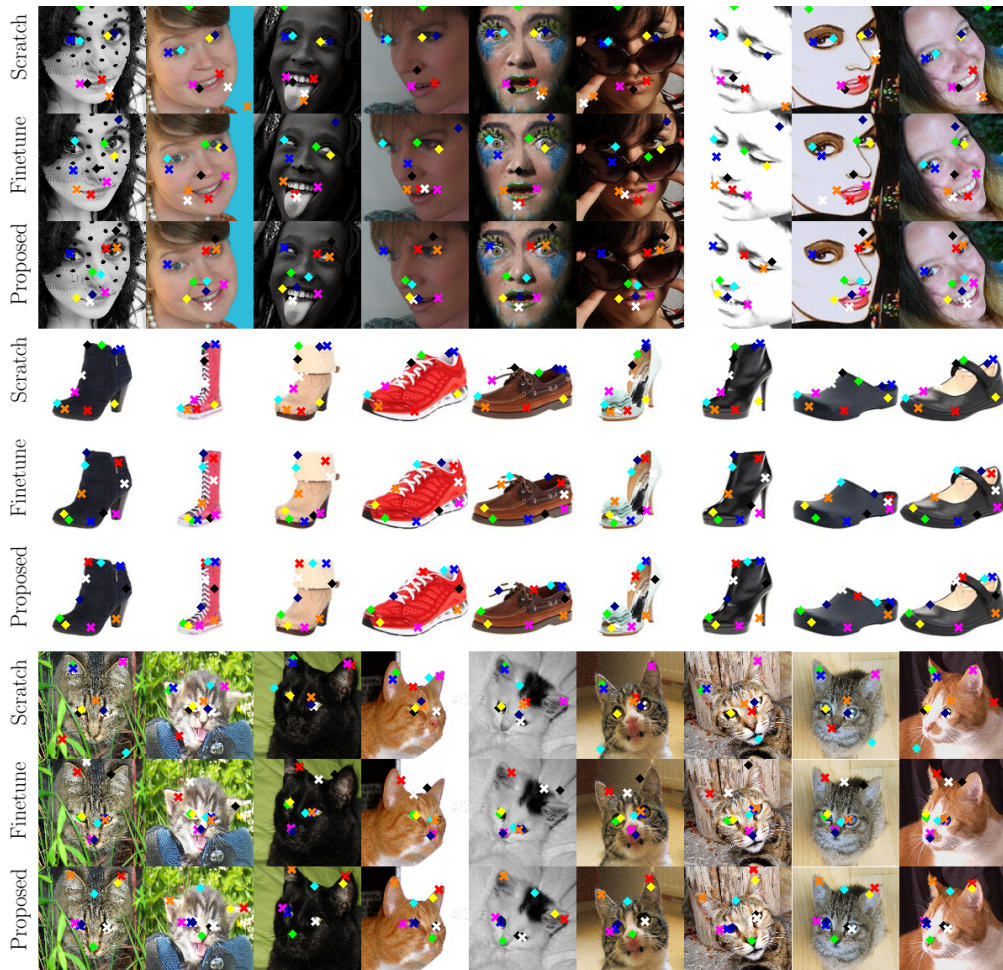

**Figure 3:** Qualitative results on AFLW, Shoes, and Cats datasets. Our method produces the most stable landmarks which is visually more evident for the faces and cats datasets. See text for detailed discussion.

| | | 1 | 2 | 3 | 4 | 5 | 6 | 7 | 8 | 9 | 10 | Avg. |
|---|---|---|---|---|---|---|---|---|---|---|---|---|
| # Epoch | 1 | 1.53 | 1.77 | 2.91 | 5.25 | 8.66 | 10.19 | 11.63 | 26.69 | 37.14 | 45.12 | 15.09 |
| | 5 | 1.60 | 1.64 | 1.69 | 1.70 | 1.83 | 2.11 | 6.97 | 39.36 | 41.06 | 45.28 | 14.32 |
| | 10 | 1.43 | 1.57 | 1.64 | 1.76 | 1.81 | 1.83 | 1.99 | 2.42 | 7.98 | 30.58 | 5.30 |
| | 110 | 1.46 | 1.47 | 1.47 | 1.54 | 1.65 | 1.66 | 1.92 | 2.07 | 4.97 | 6.99 | 2.52 |
| # Images | 10 | 1.47 | 2.02 | 2.02 | 2.15 | 2.26 | 2.54 | 2.73 | 3.17 | 6.32 | 6.40 | 3.11 |
| | 100 | 1.93 | 1.97 | 2.05 | 2.21 | 2.63 | 2.65 | 3.45 | 3.69 | 6.77 | 7.02 | 3.44 |
| | 1000 | 1.55 | 1.56 | 2.00 | 2.22 | 2.36 | 2.52 | 3.44 | 4.54 | 9.06 | 10.82 | 4.01 |
| | All | 1.46 | 1.47 | 1.47 | 1.54 | 1.65 | 1.66 | 1.92 | 2.07 | 4.97 | 6.99 | 2.52 |

**Table 4:** *Consistency* errors of our method produced by varying the quality of the core network (top), and the number of images used for training (bottom).

*Further investigating finetuning*: While our experiments clearly show that our approach significantly improves upon finetuning, the results of the latter indicate that it also constitutes an effective approach to unsupervised adaptation. To further explore this direction, we also studied (a) finetuning only the last layers of the core network, leaving the rest of the network frozen, and (b) finetuning after having learned the projection matrix. In both cases, we found that the results were slightly worse than the finetuning results reported in Table 2.

*Generalization*: The experiments throughout this paper used as core a powerful network trained on MPII [2] which has learned rich features that can serve as basis for adaptation. In addition to that, we also investigated whether our approach would also work for the inverse case, e.g. by training a core network for facial landmark localization and adapting it to detect body landmarks. In Fig. 5, we show some qualitative results of a network trained to discover 10 points on the BBC-Pose dataset [4], using as core a network trained to detect 68 landmarks on the 300W-LP dataset [47]. A detailed evaluation of each method for this challenging setting can be found in the Supplementary Material.

# 5    Conclusions

In this paper, we showed that the unsupervised discovery of object landmarks can benefit significantly from inheriting the knowledge acquired by a network trained in a fully supervised way for a different object category. Using an incremental domain adaptation approach, we showed how to transfer the knowledge of a network trained in a supervised way for the task of human pose estimation in order to learn to discover landmarks on faces, shoes, and cats in an unsupervised way. When trained under the same conditions, our experiments showed a consistent improvement of our method w.r.t. training the model from scratch, as well as fine-tuning the model on the target object category.

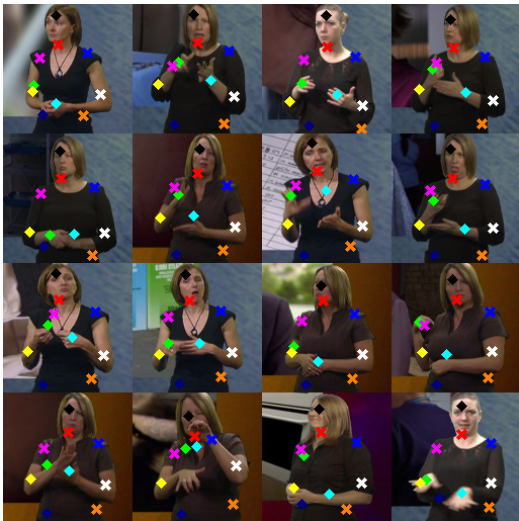

**Figure 4:** Examples of 10 discovered landmarks on BBC-Pose (test).

| | | | # Epoch | | |
|---|---|---|---|---|---|
| | $n_{im}$ | 1 | 5 | 10 | 110 |
| Fwd | 10 | 16.27 | 16.45 | 12.60 | 11.09 |
| | 100 | 11.53 | 9.43 | 8.40 | 7.64 |
| | All | 9.68 | 7.76 | 6.93 | 6.69 |
| Bwd | 10 | 43.75 | 40.75 | 17.35 | 10.05 |
| | 100 | 34.69 | 33.96 | 14.44 | 9.69 |
| | All | 33.72 | 32.86 | 13.58 | 9.19 |
| | | | # Images | | |
| | $n_{im}$ | 10 | 100 | 1000 | All |
| Fwd | 10 | 16.55 | 11.31 | 11.25 | 11.09 |
| | 100 | 13.21 | 8.22 | 8.34 | 7.64 |
| | All | 12.01 | 7.25 | 6.86 | 6.69 |
| Bwd | 10 | 16.55 | 14.58 | 15.32 | 10.05 |
| | 100 | 7.64 | 11.54 | 12.16 | 9.69 |
| | All | 7.31 | 10.77 | 11.30 | 9.19 |

**Table 5:** *Forward* and *backward* errors of our method produced by varying the quality of the core network (top), and the number of images used for training (bottom).

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
