[Supplementary Material]

# Supplementary Material
# Object landmark discovery through unsupervised adaptation

**Enrique Sanchez**[1]

[1] Samsung AI Centre
Cambridge, UK
{e.lozano, georgios.t}@samsung.com

**Georgios Tzimiropoulos**[1,2]

[2] Computer Vision Lab
University of Nottingham, UK
yorgos.tzimiropoulos@nottingham.ac.uk

## Face → Body

This Section details the experimental results obtained from adapting a core trained to detect facial landmarks in a supervised manner to discover body landmarks in an unsupervised way. The core network is trained to detect $68$ landmarks on the 300W-LP dataset [2], following the supervised setting described in Section 3. Then, the network is adapted to discover 10 keypoints on the BBC-Pose dataset [1]. The BBC-Pose contains 20 one-hour long videos of sign-language signers, with a test set comprising 1000 images. We used the official training/test partitions, and trained our network using the 10 videos belonging to the training partition. In order to deal with pose changes, we used as input to the generator a randomly chosen frame within a window of $\pm100$ frames from that used as input to the detector. We also used the random similarity transform described in Section 4 for both images. Finally, we randomly flipped both images, in order to cope with the fact that all videos were recorded with the subjects having a biased pose (subjects are on the right side of the video looking at the screen on their right side). The consistency errors for the three methods studied in our paper (scratch, fine-tuned and proposed) are shown in Table 1. Qualitative comparisons for the discovered landmarks are shown in Figs. 16-18.

| | | 1 | 2 | 3 | 4 | 5 | 6 | 7 | 8 | 9 | 10 | Avg. |
|---|---|---|---|---|---|---|---|---|---|---|---|---|
| **BBC-Test** | Scratch | 2.83 | 3.48 | 4.37 | 5.14 | 6.10 | 6.38 | 7.23 | 8.18 | 8.58 | 17.38 | 6.97 |
| | Finetune | 2.93 | 3.47 | 3.85 | 5.38 | 5.86 | 7.08 | 8.16 | 10.47 | 14.47 | 15.15 | 7.68 |
| | Proposed | 2.39 | 3.40 | 3.44 | 4.00 | 5.44 | 5.58 | 6.47 | 7.11 | 7.78 | 9.77 | 5.54 |

Table 1: *Consistency* errors on BBC-Test.

## Additional qualitative results

Below we show additional qualitative results accompanying Section 4 in the main paper.

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

Figure 1: Examples on AFLW

Figure 2: Examples on AFLW

Figure 3: Examples on AFLW

Figure 4: Examples on LS3D

Figure 5: Examples on LS3D

Figure 6: Examples on LS3D

Figure 7: Examples on MAFL

Figure 8: Examples on MAFL

Figure 9: Examples on MAFL

Figure 10: Examples on UT-Zappos50k

Figure 11: Examples on UT-Zappos50k

Figure 12: Examples on UT-Zappos50k

Figure 13: Examples on Cats Head

Figure 14: Examples on Cats Head

Figure 15: Examples on Cats Head

Figure 16: Examples on BBC-Pose (test)

Figure 17: Examples on BBC-Pose (test)

Figure 18: Examples on BBC-Pose (test)