[Reviews · NeurIPS 2019]

Reviewer 1



1. I found it hard to understand what the "finetuning" method referred in this paper is. L217 explains it to be a method that fine-tunes the weights of the network. Is it the same as the proposed approach (same objective function) but instead of using the weight transform kernels (L130), the original parameters of the supervised network are updated during the unsupervised adaptation process? This is an important baseline for the paper because it shows the importance of using - a) the weight transform kernel; b) keeping the parameters of the landmark model itself fixed. Please clarify this in the author response. 2. What is the performance of the method on the original MPII dataset after the unsupervised domain adaptation process? The authors allude to catastrophic forgetting being a motivation to use the weight transform kernels, but do not show if it actually helps. For such an experiment, it is important to show what happens if a) the weight transform kernels are used on MPII; b) the weight transform kernels are set to identity on MPII. 3. I find the analysis in this paper a bit lacking. For example, rather than the extremes - use the weight transform kernels OR finetune all weights (assuming my understanding in 1 is correct), the authors do not show any approach in between. How about a model that optimizes the same loss as in L170, but does not use the weight transform kernel and only finetunes the last few layers? Although this approach may have catastrophic forgetting, it is a good baseline to demonstrate why the particular design decisions in the paper matter. 4. What are the training hyperparameters for the scratch network? Is it trained for longer than the other methods? With enough data augmentation (example affine transforms of images)?

Reviewer 2



Given an image set of which images contain a single target category object at near the center, the submission presents a neural network method that can discover a fixed number of landmarks in an unsupervised way. The basic idea of this work is to leverage the learned knowledge of the source network pre-trained with a source object category to discover the landmarks of a target object category. To this end, the proposed landmark estimation network is designed in a simplified way of Progressive networks [29]. All the activations obtained from each layer of the pre-trained fixed source network are transformed by respective 1x1 convolution layers (i.e., linear transform in Eq. (1)). The last convolution layer is modified to output the pre-defined number of the landmark for the target object. Then, all the linear transforms are trained by backpropagation with the unsupervised task loss, landmark conditional image generation [13], on the target category data. The number of parameters is significantly lower than training the whole network from scratch, because only the linear feature transforms are trained while fixing the source network. The proposed model is largely based on [13], so the authors reproduced [13] and used it as the baselines, called Scratch and Fine-tuning. The authors showed the proposed method outperforms the fine-tuning baseline. The authors' hypothesis is due to fewer degrees of freedom by the small number of parameters to optimize than [13]. The paper is written clearly, and the experiment shows the effectiveness of the proposed method. However, the authors fail to provide understandings of the proposed algorithm's behavior and failure cases that would contribute a lot to get informative takes. Due to this missing piece, it is a bit vogue where the performance gain actually comes from and how stable the algorithm is. - Strong assumption: the proposed approach is based on a strong assumption that useful intermediate activations for the target object can be linearly spanned from other categories activation; in other words, for a target object, there exist effective convolution filters constructed by the linear combination of the specifically trained filters for a source object. The authors overlooked mentioning this point. - Category dependency: Also, there could be some other source-target cases that may fail to represent each other by the linear combination. However, the authors failed to demonstrate the effects according to the categories. Currently, the source network is trained for the human pose (16 keypoints), and the target network is adapted to different object categories. Most of the evaluations are done on the "human=>face" case. The cases of "human=>{cat head, shoes}" are only demonstrated by consistency measure in Table 3 and Figure 3, which does not truly assess the landmark accuracy (will be discussed further in the below comment). These only show very limited generalization across the category. - Taxonomy; as mentioned above, there must be effective relationships between different category pairs. Showing varying performance and knowledge transferability would be critical evaluation in this submission due to its technical contribution point on adaptation. - The proposed method requires to have a pre-trained landmark detector for a different object category a priori, which is pre-trained in a fully supervised way. Thus, the system would largely depend on the quality of the core network. While this is the case, the authors do not provide any study to understand dependency. The comments in sections would be relevant to get better understand the proposed method. Please see . - L250-251 and Table 2; I found that the authors' interpretation of these lines is good. But if the 3D rotation of the landmark in the LS3D dataset is the factor that cannot be handled by all the methods, then the LS3D errors reported in Table 2 is not reliable to be interpreted because large dominant outliers will dominantly contribute to errors. Thus, what we can parse from the reported error is that the models just do not work. So, in this case, it would be more interpretable to measure the errors only for visible landmarks, rather than measuring overall landmarks. - Landmark consistency against random similarity transform is interesting, but not strong evidence of outperforming. For example, in an extreme case, if the network always produces average landmarks regardless of any input, then it will have the perfect consistency. Thus, this metric alone cannot be used to argue accurate landmark detection.

Reviewer 3



The paper proposes a method to adapt an existing landmark detector using an unsupervised objective to discover landmarks on new categories of objects. In particular, It uses a projection matrix to adapt the network. Experiments are performed on the face, cat head, and shoe datasets. The idea of using the knowledge in an existing landmark detection is interesting, and the experiments show that the method works to some extent. However, the technical novelty of this paper is not very significant. It is a relatively straightforward combination of existing unsupervised landmark learning and domain adaptation methods. The domain adaptation is done in a general way, which is not specific to the landmark discovery problem. A relevant question (not a con itself): is it possible to adapt an image classification or object detection model to unsupervised landmark learning? In Table 1, the proposed method is not consistently better than the previous methods and the baselines. It is discouraging that using a pretrained network led to worse performance than the previous work, who trained the model from scratch. In the remaining of the experiments, comparisons are done only with the two baselines. For example, the consistency metric (which is interesting) is not tested on previous methods. It will be interesting to see fine-tuning with proper learning rate after matrix projection. It will also be interesting different how different pretrained network can impact the final landmark discovery performance.

[Author Response · NeurIPS 2019]

| # Epochs | Con. | Fwd | Bwd |
| --- | --- | --- | --- |
| 1 | 15.09 | 9.68 | 33.72 |
| 5 | 14.32 | 7.76 | 32.86 |
| 10 | 5.30 | 6.93 | 13.58 |
| 110 (paper) | 2.52 | 6.69 | 9.19 |

| # imgs | Con. | Fwd | Bwd |
| --- | --- | --- | --- |
| 10 | 3.11 | 12.01 | 7.31 |
| 100 | 3.44 | 7.25 | 10.77 |
| 1000 | 4.01 | 6.86 | 11.30 |
| All (paper) | 2.52 | 6.69 | 9.19 |

Table 1: *Con.*: Avg. Consistency error, *Fwd*: Forward error, *Bwd*: Backward error. **Left**: results using different core networks (GC1). **Right**: results by varying the # of training images (GC2).

● **All Rs:** Thank you for your insightful comments. There were some concerns raised regarding conducting additional experiments which we hope are addressed in General Comments (GC) 1 & 2 and individual responses.

● **GC1: Accuracy vs quality of core network** While our core (fully supervised) network in the paper was trained for 110 epochs (PCKh=79%), we also used the weights after the 1st (PCKh=22.95%), 5th (PCKh=57.67%), and 10th (PCKh=57.67%) epochs of the training process. With these networks as cores, the achieved accuracies are shown in Table 1 (**Left**). Due to limited space, we report only on AFLW dataset using the full dataset to train the regressor (see also Sec 4.2 of the main paper). These results clearly illustrate the importance of training a powerful core network.

● **GC2: Accuracy vs # training images** To this end, we chose a random subset of 10, 100, and 1000 images. The achieved accuracies are shown in Table 1 (**Right**). Visually, we observed that using only 10 images makes the network converge to a singe point, hence the poor forward and good backward and consistency errors. Using a subset of 1000 images, results are getting close to using the whole CelebA dataset ( 200k images). Fine-tuning the core network using 1000 images, yields consistency, forward and backward errors of 5.38, 7.52, and 13.36 which are much worse than our approach. This shows that our method is more effective with limited training data. We will include an in-depth study.

● **R1**: ● **R1.1**: *1. Finetune*: "Finetuning" is the same as training from scratch but with the weights initialized from the human pose estimation network. Thank you, we will clarify this. ● **R1.2**: *2. Performance on MPII*: On MPII, one can just use the core network so no drop in performance occurs. This is also equivalent to setting the weight transform kernels to identity. If we understood correctly, you also requested to see what happens if the target domain is also set to MPII. In this case, the discovered landmarks are different from the ones that the core network learns to predict in a supervised manner. This is not unreasonable as the objective functions for the supervised and the unsupervised cases are completely different. Thank you for this we will include it. ● **R1.3**: *3. Fine-tuning only the last layers*: We did try this: the results on AFLW for the consistency, forward, and backward errors are 4.10, 7.6, and 12.0 showing that this is actually worse than fine-tuning the whole network. ● **R1.4**: *4. Hyperparams*: All networks are trained in the same way (with augmentation) until the predicted points on the training set do not change w.r.t a threshold.

● **R2**: ● **R2.1**: *Strong assumption + category dependency*: Indeed, the target domain learned filters are a linear combination of the core ones; however this doesn't seem to be so restrictive as long as the core network is trained on a difficult task (this is why we chose the one of human pose estimation). Thank you for this, we will also show the inverse experiment (face->human) in the revised version. So far we have tried face->MPII with little success for all methods probably because MPII is too hard (e.g. multiple objects per image). ● **R2.2**: *Evaluation for cat head, shoes.*: We follow prior works (e.g. [13]) and evaluate shoes and cats qualitatively. On top of that we report consistency measure. Please see supplementary. We believe this is sufficient. ● **R2.3**: *Taxonomy.*: This is interesting but given the lack of annotated data it is hard to conduct. In any case, the most critical is to have a core network trained on a difficult task, see also R2.1. ● **R2.4**: *Interpretation of 3D data*: Our method does work for LS3D as both qualitative results and the consistency measure show (see Table 3, paper). However, as the forward error shows, it is hard to learn the mapping between 2D landmarks (as learned by our method) and the 3D landmark annotations. Unfortunately, we do not have visibility labels so we cannot measure accuracy on visible landmarks only. Thank you for this we will clarify it. ● **R2.5**: *Landmark consistency*: We agree that the consistency alone is not sufficient. However, as we show in our work, forward error is not sufficient either. This is why we use ALL 3 errors (forward, backward, and consistency) always in combination with visual quality. We believe that such evaluations are SOTA. ● **R2.6**: *Performance according to quality of core*: Please see GC1. ● **R2.7**: *Performance on small no. of training images*: Please see GC2. ● **R2.8**: *Categorical dependency relationships*: We will also include face->human. See R2.1 and R2.3.

● **R3**: ● **R3.1**: *Technical contribution*: We are the first to propose the mixed training strategy of Fig 1c (core–supervised, target domain–unsupervised) and show that this is beneficial for constraining the optimization problems encountered in unsupervised landmark discovery. ● **R3.2**: *Table 1*: In all tables including Table 1, our method always outperforms both trained-from-scratch and fine-tune networks when these are implemented in-house hence ensuring a fair comparison. The implementations of [13] and [45] report better numbers but they use different ways to process and crop the images which can significantly impact the results. Also please notice that these numbers are the forward errors which as we show in our paper can be biased. Adapting an image classification model could be possible, but we opted for using a human pose one since this is readily available and we are interested in landmarks. ● **R3.3**: *Comparisons with only two baselines.*: One of our baselines is the trained-from-scratch network from [13] which is SOTA. The other baseline improves upon that. Hence, we have ensured sufficient comparison with SOTA. ● **R3.4**: *Fine-tuning after matrix projection*: We tried this and observed no further improvement: we got 2.51, 6.46, and 9.52 for the consistency, forward, and backward errors, respectively. We will add this to the paper. ● **R3.5**: *Try other core nets*: Please see GC1.

[Meta-Review · NeurIPS 2019]

This paper received mixed reviews, with two reviewers in favor and one reviewer against. The main point of discussion among reviewers was due to the novelty. After considering the rebuttal, this concern was clarified.